# POLDIP3: At the Crossroad of RNA and DNA Metabolism

**DOI:** 10.3390/genes13111921

**Published:** 2022-10-22

**Authors:** Manrose Singh, Sufang Zhang, Alexis M. Perez, Ernest Y. C. Lee, Marietta Y. W. T. Lee, Dong Zhang

**Affiliations:** 1Department of Biomedical Sciences, College of Osteopathic Medicine, New York Institute of Technology, Northern Blvd, Old Westbury, NY 11568, USA; 2Department Biochemistry and Molecular Biology, New York Medical College, Valhalla, NY 10595, USA

**Keywords:** POLDIP3, DNA replication, DNA damage response, R-loops, transcription, pre-mRNA splicing, mRNA export, translation

## Abstract

POLDIP3 was initially identified as a DNA polymerase delta (Pol δ) interacting protein almost twenty years ago. Intriguingly, it also interacts with proteins involved in a variety of RNA related biological processes, such as transcription, pre-mRNA splicing, mRNA export, and translation. Studies in recent years revealed that POLDIP3 also plays critical roles in disassembling genome wide R-loop formation and activating the DNA damage checkpoint in vivo. Here, we review the functions of POLDIP3 in various RNA and DNA related cellular processes. We then propose a unified model to illustrate how POLDIP3 plays such a versatile role at the crossroad of the RNA and DNA metabolism.

## 1. Introduction

More than thirty years ago, two labs, including ours, independently cloned the gene encoding the catalytic subunit of DNA polymerase delta (δ), POLD1/p125 [1,2,3]. The detailed history of how the holoenzyme of DNA polymerase δ, or Pol δ, which includes four different subunits (POLD1/p125, POLD2/p50, POLD3/p68, and POLD4/p12), was identified has been reviewed previously [4]. Many studies in the past thirty years have demonstrated that Pol δ plays a critical role in DNA replication and DNA repair, including synthesis of the lagging strand during S phase [4]. POLDIP3, also known as DNA polymerase delta interacting protein 46 (PDIP46) or DNA polymerase delta interacting protein 3 (PDIP3), was initially identified as a Pol δ-associated protein through a two-hybrid screening [5]. Soon after, it was re-identified by Richardson and colleagues through another two-hybrid screening as a binding partner and substrate of S6K1, which is a key downstream target of mTOR and PI3K and implicated in the regulation of cell and organism growth [6]. Therefore, they gave it another name, SKAR, for the S6K1 Aly/REF-like target. Subsequently, it was a positive hit again through a third two-hybrid screening as an interacting protein of human Enhancer of Rudimentary Homolog (ERH), which plays a role in transcription regulation [7]. Furthermore, it was identified in a proteomic screening to be associated with the TREX complex, which is involved in transcription, mRNA processing, and mRNA export [8,9]. Given the diverse binding partners of POLDIP3 and the range of cellular processes involved, examining the existing data on POLDIP3 is an intriguing prospect. Here, we review the reported in vitro and in vivo functions of POLDIP3 in RNA-related, as well as DNA-related biological processes. We then propose a unified model to illustrate how POLDIP3 coordinates and integrates these diverse cellular processes.

## 2. Domains, Binding Partners, and Modifications of POLDIP3

Multiple groups, including ours, have detected two different POLDIP3 protein variants, POLDIP3α and POLDIP3β, with a size of 421 amino acids and 392 amino acids, respectively, in multiple human cell lines (Figure 1A) [6,9,10]. POLDIP3α contains two recognizable domains. The first domain contains five AlkB homolog 2 PCNA Interacting Motifs (APIM) located between arginine-53 and lysine-124 [10]. The APIM is a PCNA-binding motif with the consensus sequence of K/R-F/Y/W-L/I/V/A-L/I/V/A-K/R and can interact with the post-translationally modified (PTM) PCNA [11], which includes sumoylation and ubiquitination [12]. In addition to binding to PCNA, this region also binds to the POLD2/p50 subunit of Polδ [10]. The second domain contains an RNA Recognition Motif (RRM) located between threonine-280 and asparagine-351 (Figure 1A). This RRM belongs to the Aly/REF family of RNA binding proteins, which can bind modified RNA, such as 5-methylcytosine in the coding regions of mRNAs [13].

POLDIP3 interacts with a variety of proteins that are important for multiple RNA- and DNA-related biological processes. The experimentally validated ones are summarized in Figure 1B,C. At present, there is no crystal structure available for POLDIP3. Figure 2 depicts the predicted three-dimensional (3D) structure of human POLDIP3 using the Alphafold, which suggests that other than the well-folded RRM domain, most of POLDIP3 are highly disordered and flexible. Based on the data currently available from the GenBank, the homologs of human POLDIP3 can be found in chimpanzee, dog, cow, mouse, rat, chicken, and frog, but not in the lower organisms, such as flies, worms, and yeasts (Figure 3).

POLDIP3 protein can be modified by phosphorylation under various conditions (Figure 1A and Table 1). For example, DNA-damaging agents induced phosphorylation at serine-42 [14]. In addition, two proteomic studies have shown that POLDIP3 can also be modified by sumoylation at multiple lysine residues [15,16]. POLDIP3 contains at least three SUMO consensus sequences (Figure 4A) [17]. Most importantly, we experimentally confirmed that POLDIP3 can indeed be sumoylated using both in vitro and in vivo sumoylation assays (Figure 4B,C). Some of the sumoylation events occurred in response to replication stress, indicating potentially specialized roles for POLDIP3 in replication stress response that need further investigation [16].

## 3. POLDIP3 Functions in a Variety of RNA-Related Cellular Processes

As mentioned above, two separate two-hybrid screenings identified POLDIP3 as a major binding partner for S6K1 and ERH, respectively [6,7]. S6K1 is an important downstream target of mTOR and plays a critical role in regulating protein translation and cell size [18]. In response to insulin stimulation, Richardson and colleagues showed that the hyper-phosphorylated S6K1 binds POLDIP3 and phosphorylates it at serine-383 and serine-385 both in vitro and in vivo to regulate the cell size [6]. Later, the same group showed that POLDIP3 is deposited at the exon junction during pre-mRNA splicing and becomes part of the exon junction complex (EJC), which then facilitates the recruitment of S6K1 to the spliced RNA to boost the translation efficiency of the spliced mRNA [19]. One of the most versatile protein complexes involved in various processes of RNA metabolism is called the TREX complex. The TREX complex plays an important role in transcription, mRNA processing, as well as mRNA export [20]. Consistent with its potential role in RNA metabolism, Folco and colleagues found that POLDIP3 also associates with the TREX complex in an ATP-dependent manner, and overexpression of POLDIP3 induces retention of the polyA+ RNA in nuclear speckles [8,9]. Further supporting the role of POLDIP3 in translation, Kroczynska and colleagues showed that, in addition to S6K1, POLDIP3 can also be phosphorylated by p90 ribosomal protein S6 kinase (RSK) in response to IFN-α activation [21]. The phosphorylated POLDIP3 then stimulates the interaction between the eukaryotic initiation factor, eIF4G, and the CBP80 immune complexes and recruits them to the 5′ 7-methylguanosine cap of mRNA for protein translation.

ERH functions as a transcription repressor and as a known modulator of alternative splicing in a variety of biological processes [22]. Most intriguingly, ERH was also identified in a whole-genome siRNA screening for factors that regulate replication stress-induced DNA damage and DNA synthesis [23]. Among the many transcriptional targets of ERH, there are a variety of factors involved in DNA replication and DNA damage response (DDR) [23,24,25]. It was further demonstrated that ERH also regulates the replication stress response by modulating the alternative splicing of ATR, a central player in DDR [26,27]. Whether the interaction between ERH and POLDIP3 is essential for the functions of ERH in DNA replication and DDR requires further investigation.

## 4. POLDIP3 Plays an Important Role in a Variety of DNA Related Cellular Processes

### 4.1. POLDIP3 Robustly Stimulates the Enzymatic Activity of DNA Polymerase δ

DNA replication is an essential cellular process requiring tight regulation, as mistakes made during DNA replication can contribute to the development of several human diseases, notably cancer. For example, based on the correlations between the risk of developing tumors for various tissues and the number of cell divisions that the tissue-specific stem cells likely undergo in their lifetime, Tomassetti and colleagues proposed that the DNA replication errors may account for about two-thirds of cancer incidence [28,29]. Different organisms have developed elaborate strategies to maximize the fidelity of DNA replication [30,31,32,33]. In eukaryotes, DNA replication during S phase is performed by the coordinated actions of three DNA polymerases: DNA polymerase α (Pol α), DNA polymerase delta (Pol δ), and DNA polymerase epsilon (Pol ε) [33]. Pol α synthesizes short RNA primers to initiate DNA synthesis. Subsequently, Pol ε extends the leading strand continuously. Pol δ extends the lagging strand by synthesizing and processing of the Okazaki fragments [31].

Pol δ (or Pol δ4) is made up of four core components: POLD1/p125, POLD2/p50, POLD3/p68, and POLD4/p12. POLD1/p125 is the catalytic subunit of Pol δ, while POLD2/p50, POLD3/p68, and POLD4/p12 play various important structural and regulatory roles. To further elucidate the functions of Pol δ and identify novel regulators of Pol δ, we performed a two-hybrid screen almost twenty years ago using POLD2/p50 as the bait [5]. POLDIP3 was identified as one of the POLD2/p50-interacting proteins. Subsequently, using purified recombinant proteins, we demonstrated a direct interaction between POLDIP3 and POLD2/p50. Additionally, we mapped the regions of their interactions: between amino acids 71 and 141 in POLDIP3, which overlaps with its APIM domain, and between amino acids 252 and 400 in POLD2/p50. Furthermore, we found that POLDIP3 also directly binds PCNA via amino acids 1 to141, where its APIM motifs reside. As mentioned above, the APIM motifs primarily bind to the modified PCNA. It has been recognized that, in response to replication stress, PCNA can undergo a variety of modifications, including mono-ubiquitination, poly-ubiquitination, and sumoylation [12]. These modifications likely regulate the interactions between POLDIP3 and PCNA in vivo.

Using a singly primed 7.2 kb M13 DNA as the in vitro primer extension substrate, we demonstrated that POLDIP3 could robustly stimulate the processivity of Pol δ4 in the presence of unmodified PCNA. When using a shorter DNA oligonucleotide (~70 nucleotides) as the in vitro primer extension substrate, we showed that both POLDIP3-WT and POLDIP3-ΔRRM, but not the POLDIP3-APIM mutant, can robustly stimulate the DNA synthesis activity of Pol δ4 in the absence of PCNA. However, the stimulating effect is less robust on Pol δ3, which consists of only POLD1/p125, POLD2/p50, and POLD3/p68, suggesting that Pol δ4 is likely the primary regulatory target of POLDIP3. Additionally, using the same DNA oligonucleotide substrate, we also showed that POLDIP3-WT, but not the POLDIP3-APIM mutant, is able to stimulate the strand displacement activity of Pol δ4. Most intriguingly, we showed that both POLDIP3-WT and POLDIP3-ΔRRM stimulate primer extension by Pol δ4 through a template with a hairpin, suggesting that POLDIP3 may be required for Pol δ4 to overcome various DNA replication barriers and alleviate replication stress [10]. In this regard, there is a growing appreciation of Pol δ4 in replication stress response and DNA repair [34].

Collectively, our comprehensive in vitro biochemical studies strongly established POLDIP3 as an important activator of Pol δ4 and, to a lesser extent, Pol δ3, which requires its APIM domain but not its RRM domain.

### 4.2. POLDIP3 and RTEL1 Facilitate the Disruption of R-Loop Formation

Regulator of telomere length 1 (RTEL1) is an important DNA helicase and has been implicated in the disruption of a variety of DNA secondary structures to facilitate DNA replication and DNA repair [35]. In a recent proteomic study, Bjorkman and colleagues identified POLDIP3 as a RTEL1-associated protein in vivo [36]. Using purified recombinant proteins, they also established a direct physical interaction between RTEL1 and POLDIP3 in vitro. They further showed that, in response to replication stress induced by camptothecin, a DNA topoisomerase I inhibitor, RTEL1 and POLDIP3 are recruited to chromatin in a mutually dependent manner. Most importantly, they showed that R-loops, which are hybrid molecules formed via complementary base pairing between DNA and RNA, accumulate in RTEL1- or POLDIP3-deficient U2OS, a commonly used cell line for investigating the Alternative Lengthening of Telomeres (ALT) pathway, suggesting that the RTEL1-POLDIP3 may facilitate the disruption of R-loops, which are known to hinder the progression of the DNA replication machinery [37]. In a separate study, Tumini and colleagues showed that the depletion of POLD3/p68 induces R-loop-dependent DNA damage in HeLa cells [38]. However, the authors failed to detect R-loop accumulation in POLD3/p68 or POLD1/p125 deficient cells, suggesting that Pol δ is only involved in the repair/re-start of the stalled replication forks caused by the R-loop accumulation.

Taken together, these recent studies suggest that RTEL1-POLDIP3-Pol δ play an important role in disrupting the R-loops and facilitating the repair or restart of stalled forks induced by R-loop accumulations.

### 4.3. POLDIP3 Activates and Maintains the DNA Damage Checkpoint

To further investigate the functions of POLDIP3 in vivo, we generated POLDIP3 genetic knockout (KO) using CRISPR/Cas9 technology in multiple cancer cell lines [39]. We demonstrated that the POLDIP3-KO cells show an increased sensitivity to replication blockers compared to the wild-type (WT) cells (POLDIP3-WT). In addition, we also detected a strong interaction between POLDIP3 and RPA and between POLDIP3 and Tipin in vivo, two proteins that are known to be involved in DNA replication and DDR. Most importantly, we demonstrated that POLDIP3-KO cells are defective both for the initial activation of the DNA damage checkpoint and the later maintenance of the DNA damage checkpoint over an extended period of time. Furthermore, we showed that POLDIP3 can be actively recruited to the telomeres in the ALT cells that are experiencing replication stress and facilitate the activation of DNA damage checkpoint at the ALT telomeres [40]. Consistent with these newly discovered checkpoint functions, Matsuoka and colleagues reported that, in response to DNA damage, POLDIP3 can be phosphorylated at serine-42 (Figure 1), which is a potential target of ATM and ATR [14]. In addition, Blasius and colleagues identified two potential Chk1 phosphorylation sites in POLDIP3 (serine-42 and serine-63, Figure 1) [41]. However, whether these phosphorylation events directly contribute to the checkpoint function of POLDIP3 requires further investigation.

### 4.4. POLDIP3 Associates with Telomeres and Facilitates Replication Stress-Induced Checkpoint Activation

Growing evidence suggests that POLDIP3 likely also plays an important role at telomeres, especially in the ALT cells. Human telomeres are protected by a six-protein complex, called Shelterin, which consists of TRF1, TRF2, Rap1, TIN2, TPP1 and POT1 [42]. In a proteomic study of Shelterin-associated proteins, Giannone and colleagues showed that POLDIP3 co-purifies with both TRF2 and POT1 [43]. In a separate proteomic study, Garcia-Exposito and colleagues identified POLDIP3 as a potential TRF1-associated protein [44]. Intriguingly, they found that the POLDIP3-TRF1 interaction is more robust in the ALT cells than in the telomerase positive cells. Consistent with these proteomic studies, we demonstrated that POLDIP3 is recruited to the ALT telomeres that experience replication stress [39]. Most importantly, we showed that POLDIP3 facilitates the replication stress-induced DNA damage checkpoint activation [39]. Furthermore, there are also telomere-specific R-loops, namely, the telomere repeat containing RNA (TERRA) R-loops, which are known to be more enriched at the ALT telomeres [45]. Most recently, we and others have shown that the accumulation of TERRA R-loops induces robust replication stress at the ALT telomeres [46,47]. We thus speculate that the RTEL1 and POLDIP3 interaction may also be important for the regulation of TERRA R-loops at the ALT telomeres. Collectively, these studies indicate that POLDIP3 plays a crucial role in the replication stress response at telomeres.

## 5. A Unified Model of POLDIP3 in Various RNA- and DNA-Related Cellular Processes

It has been almost twenty years since we initially identified POLDIP3 as a Pol δ interacting protein. Studies since then have highlighted POLDIP3 as an important intermediary of a variety of RNA- and DNA-related biological processes, including transcription, pre-mRNA splicing, mRNA export, protein translation, DNA replication, DNA replication stress response, and DNA damage checkpoint activation. To illustrate the multi-faceted functions of POLDIP3, we propose the following model (Figure 5).

At a certain point during transcription, POLDIP3 is recruited to the nascent RNA by a yet-to-be-identified mechanism. During the transcription of certain genes, for example, those that are involved in DNA replication, DNA repair and DDR, POLDIP3 interacts with ERH and regulates their transcriptional activities. Through its interaction with ERH, POLDIP3 also modulates the alternative splicing of various pre-mRNAs, which include many of those that are involved in DNA replication, DNA repair, and DDR. Subsequently, POLDIP3 recruits the TREX complex and facilitates the export of mRNA into the cytosol for translation. During translation, S6K1 and/or RSK phosphorylate POLDIP3 and promote its interaction with eIF4G and the CBP80 immune complex to preferentially translate those spliced mRNA in various tissues and cell types, and/or induced by certain stress conditions, for example, DNA replication stress.

In the face of transcription-replication conflicts (CRT) and/or paused forks due to abnormal accumulation of R-loops, since it is already situated in the vicinity of the active transcription sites, POLDIP3 interacts with RPA and Tipin to activate and maintain the replication stress checkpoint. At the same time, it can also interact with RTEL1 in order to disassemble the accumulated R-loops. In the case that the CRT and/or accumulated R-loops cannot be resolved on time, which then leads to the stalling or collapsing of DNA replication fork, POLDIP3 then recognizes and binds the modified PCNA, and then recruits and stimulates the activity of Pol δ during the repairing and re-starting of the stalled replication fork.

Although significant progress has been made in uncovering the biological functions of POLDIP3, many important questions remain to be answered. For example, when and how is POLDIP3 recruited to the nascent RNA? As shown in Figure 1 and Table 1, POLDIP3 can be modified by phosphorylation as well as sumoylation at many sites. How do these modifications affect the biological functions of POLDIP3 in the RNA and DNA related cellular processes? What are the functions of POLDIP3 related to the physiology and disease development in mammals? We are confident that there will be more exciting new discoveries of POLDIP3 in the next twenty years.

## Figures and Tables

**Figure 1 genes-13-01921-f001:**
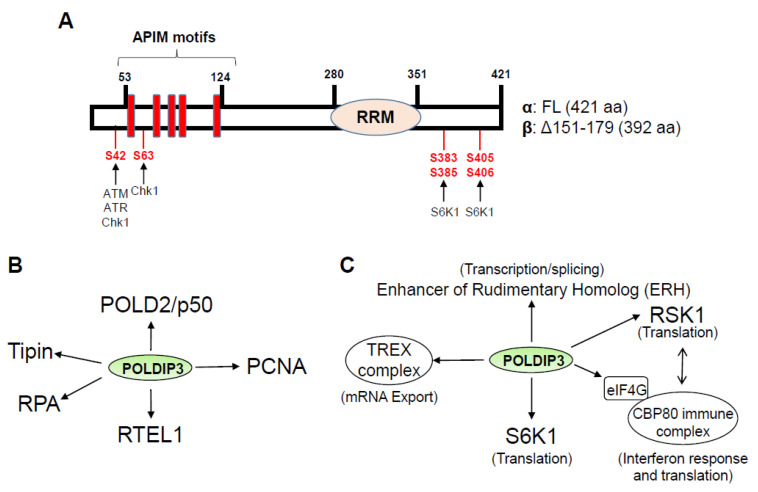
(**A**) Domain structure of human POLDIP3. The full-length POLDIP3 protein (POLDIP3-FL or POLDIP3α) contains 421 amino acids (aa). The β variant of POLDIP3 protein (POLDIP3β) contains 392 aa with residues between 151 and 179 deleted from the POLDIP3α. The five APIM motifs, illustrated as five red rectangles, span from arginine-53 to lysine-124. The RRM domain spans from threonine-280 to asparagine-351. The serine residues and the potential kinases that phosphorylate them under different conditions are also highlighted. (**B**) POLDIP3 interactors that are implicated in DNA metabolism. (**C**) POLDIP3 interactors that are implicated in RNA metabolism.

**Figure 2 genes-13-01921-f002:**
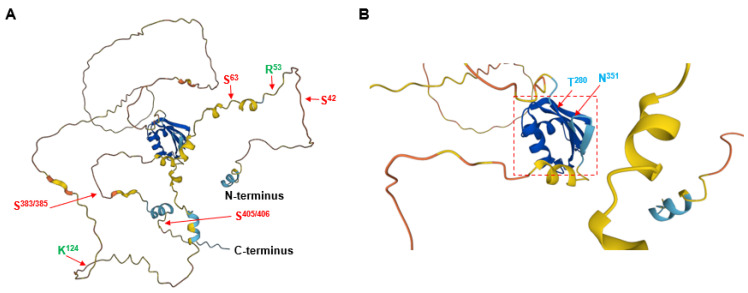
Predicted three-dimensional (3D) structure of human POLDIP3 using the AlphaFold Protein Structure Database (https://alphafold.ebi.ac.uk/, accessed on 19 October 2022). (**A**) The 3D structure of the full-length protein of human POLDIP3. (**B**) The 3D structure of the RRM domain of human POLDIP3 is highlighted with a dashed red square. The two residues flanking the five APIM motifs, arginine-53 and lysine-124, are shown in GREEN colored fonts. The two residues flanking the RRM domain, threonine-280 and asparagine-351 are shown in BLUE colored fonts. The serine residues (serine-42, -63, -383/385, and -405/406) that can be phosphorylated are shown in RED colored fonts.

**Figure 3 genes-13-01921-f003:**
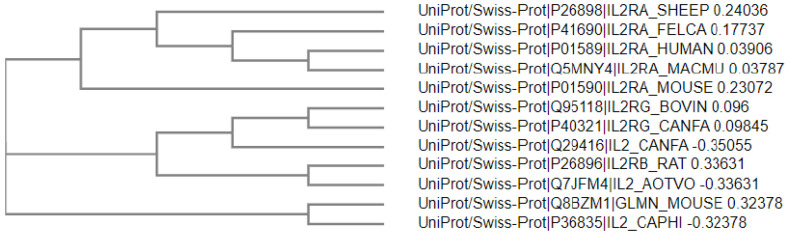
Phylogenetic tree of POLDIP3 homologs. GeneBank was used to align the different homologs of POLDIP3. The phylogenetic tree of POLDIP3 homologs was created using the Simple Phylogen from EMBL-EBI.

**Figure 4 genes-13-01921-f004:**
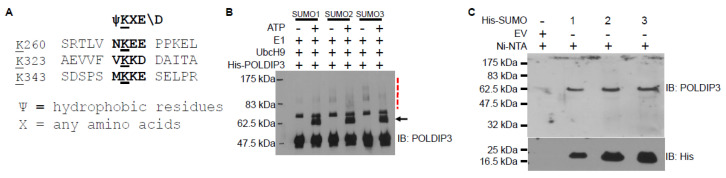
POLDIP3 can be sumoylated both in vitro and in vivo. (**A**) The SUMO consensus sequences in human POLDIP3 protein. (**B**) In vitro sumoylation assay. Purified recombinant proteins were used in the in vitro sumoylation assay in the presence (+) or absence (−) of ATP. The black arrow indicates the mono-sumoylated POLDIP3. The dashed red line marks the poly-sumoylated POLDIP3. (**C**) In vivo sumoylation assay. HEK293T cells were transfected with either empty vector (EV) or plasmids expression His-SUMO-1, or His-SUMO-2, or His-SUMO-3. Ni-NTA beads were used to pull-down the sumoylated proteins, which were then immunoblotted (IB) with antibodies as indicated on the right.

**Figure 5 genes-13-01921-f005:**
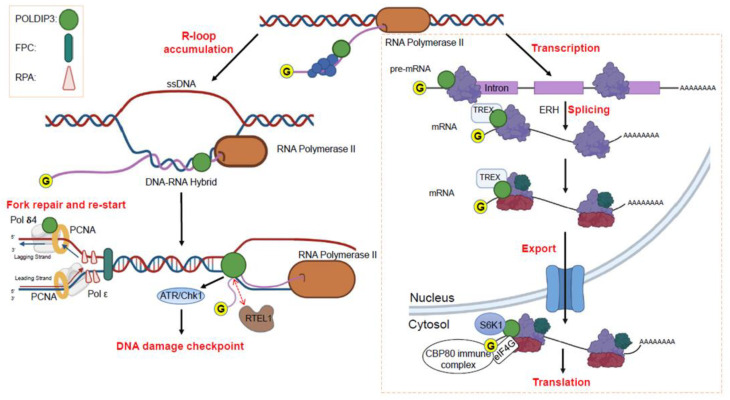
A model of how POLDIP3 orchestrates multiple biological processes related to RNA and DNA metabolism. FPC: fork protection complex that consists of Tipin, Timeless, Claspin and AND1. RPA: replication protein A that consists of RPA70, RPA34, and RPA14. TREX: transcription and export complex that consists of the THO complex and a group of accessory proteins.

**Table 1 genes-13-01921-t001:** Modifications of POLDIP3.

	Position	Sequence	Kinase/Enzyme	References
Phosphorylation	S42	QQGLLS*QSTRTA	ATM/ATR; Chk1	14 and 40
	S63	QKIGLS*DARLK	Chk1	40
	S127	SLKRSS*PAAFI		6
	S275	AEPVLS*PLEGT		6
	S383/S385	PRRVNS*AS*SSNPP	S6K1	6
	S405/S406	KALFKS*S*GASV		6
Sumoylation	K59	FDARQK*IGLSD		15
	K68	SDARLK*LGVKD		15
	K72	LKLGVK*DAREK		15
	K102	MLNSRK*QQTTV		15
	K144	VTPALK*LTKTI		15
	K147	ALKLTK*TIQVP		15
	K155	QVPQQK*AMAPL		15
	K178	NNHQAK*QNLYD		15
	K197	ASVPTK*QMKFA		15
	K200	PTKQMK*FAASG		15
	K223	KLSMSK*ALPLT		15
	K248	SSIRTK*ALTNM		15
	K260	RTLVNK*EEPPK		15
	K281	PLEGTK*MTVNN		15
	K323	EVVFVK*KDDAI		15
	K333	ITAYKK*YNNRC		16
	K372	DSPSMK*KESEL		15 and 16
	K400	PDTILK*ALFKS		15
	K418	QPTEFK*IKL		15

Notes: * mark the sites of modifications. The bolded and underlined letters mark the SUMO consensus sequences.

## Data Availability

Not applicable.

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
