# Peer review of "POLDIP3: At the Crossroad of RNA and DNA Metabolism"

_genes, 2022, doi:10.3390/genes13111921_

Round 1

Reviewer 1 Report

The authors provide a comprehensive review on POLDIP3. 

Comments:

Lane 49: The authors should extend the description of figure 1.

Lanes 78 to 82: A short description of the TREX complex should be included before talking about the association of POLDIP3 with the TREX complex.

Lanes 92/93: The multiple use of ‘many’ should be avoided.

Lanes 69, 98 and 204: The authors should comment on the titles, for example….. model of POLDIP3 (action?) in ……

Lane 311: Reference seems to be incomplete.

In general the format of the references should be revised.

Figure 2: A: Reference for the SUMO consensus sequence in POLDIP3 should be included. B: all proteins should be labeled to the right.

Figure 3: Left part of the Figure is cut off. The authors should comment if the figure is based on previous publications.

Author Response

Response to the reviewers’ comments:

We want to thank the reviewers for all their insightful comments. In the followings, we addressed their comments point-by-point.

Lane 49: The authors should extend the description of figure 1.

Response: we have extended the description of Fig 1.

Lanes 78 to 82: A short description of the TREX complex should be included before talking about the association of POLDIP3 with the TREX complex.

Response: we have added a short description of the TREX complex in the first paragraph of Section-3 before talking about the association of POLDIP3 with the TREX complex.

Lanes 92/93: The multiple use of ‘many’ should be avoided.

Response: we have changed one of the “many” to “a variety of” in the second paragraph of Section-3

Lanes 69, 98 and 204: The authors should comment on the titles, for example….. model of POLDIP3 (action?) in ……

Response: we have modified the titles as suggested by the reviewer.

Lane 311: Reference seems to be incomplete.

Response: since our paper has just been published, the EndNote has not obtain the complete information yet. We will make sure that the reference contains all the information before publication.

In general the format of the references should be revised.

Response: we have re-formated the reference using the “numbered” style from EndNote. GENE may change it later.

Figure 2 (Now Fig 4): A: Reference for the SUMO consensus sequence in POLDIP3 should be included. B: all proteins should be labeled to the right.

Response: we have added the reference in the third paragraph of section-2. We also moved all the protein labels to the right

Figure 3 (Now Fig 5): Left part of the Figure is cut off. The authors should comment if the figure is based on previous publications.

Response: Fig 3 (the new Fig 5) is created by us based on our understanding of the literatures.

Reviewer 2 Report

Singh et al. have reviewed in depth the evidence for the possible involvement of POLDIP3 in biological processes related to RNA and DNA metabolism, proposing a unified model that illustrates how POLDIP3 could participate in the coordination and integration of these biological processes. Although the proposal is interesting, the manuscript requires some corrections and improvements:

In the Introduction, it is necessary to briefly explain the importance and functions of DNA polymerase delta to better contextualize the role of POLDIP3 in DNA metabolism.

The description of possible interactions of POLDIP3 with other proteins is based primarily on two-hybrid screening experiments, which are prone to false positives. In the references given, has it been verified that the authors of these papers have taken adequate actions to limit the occurrence of false positives? As a guide, I indicate below a review containing good practices for experiments based on two-hybrid screening:

Serebriiskii, Ilya G., and Erica A. Golemis. "Two-hybrid system and false positives." Two-Hybrid Systems. Humana Press, 2001. 123-134.

- In the reference [1], POLDIP3 is called PDIP38 (neither PDIP46 nor PDIP3). This nomenclature needs to be clarified. On the other hand, in which species does this protein receive the name POLDIP3?

- Line 41: the authors talk about "aspartic acid-55", while Figure 1 shows that it is located in position 53. Which is the correct one? On the other hand, to which species does the sequence referred to in Figure 1 belong?

- Line 47: what kind of modified RNA? mRNA? lncRNA? rRNA? tRNA? microRNA? Based on the reference given, it seems to be mRNA, but it should be clearly indicated in the text. On the other hand, what kind of modification is involved? Based on the reference given, it seems to be 5-methylcytosine. Where are these modifications located? UTR? CDS?

- It should be explained what the red rectangles in Figure 1A mean.

- Figure 3 is excessively pixelated and the label "Fork repair and re-start" is cropped at the left border.

- It is necessary to include a phylogeny of POLDIP3 with other evolutionarily related proteins and discuss its evolutionary conservation in eukaryotes.

- It is necessary to include the predicted 3D structure of POLDIP3. It can be found in the AlphaFold Protein Structure Database (https://alphafold.ebi.ac.uk/). The locations of functional domains and residues that may undergo modifications should be highlighted in this structure. I recommend using a different color for each functional domain of the protein and a color code according to the type of chemical modification that the residues may undergo.

Author Response

Response to the reviewer #2’s comments:

We want to thank the reviewers for all their insightful comments. In the followings, we addressed their comments point-by-point.

Singh et al. have reviewed in depth the evidence for the possible involvement of POLDIP3 in biological processes related to RNA and DNA metabolism, proposing a unified model that illustrates how POLDIP3 could participate in the coordination and integration of these biological processes. Although the proposal is interesting, the manuscript requires some corrections and improvements:

- In the Introduction, it is necessary to briefly explain the importance and functions of DNA polymerase delta to better contextualize the role of POLDIP3 in DNA metabolism.

Response: in the introduction (section-1), we have added a few brief statements on the importance and functions of DNA polymerase delta.

- The description of possible interactions of POLDIP3 with other proteins is based primarily on two-hybrid screening experiments, which are prone to false positives. In the references given, has it been verified that the authors of these papers have taken adequate actions to limit the occurrence of false positives? As a guide, I indicate below a review containing good practices for experiments based on two-hybrid screening:

Serebriiskii, Ilya G., and Erica A. Golemis. "Two-hybrid system and false positives." Two-Hybrid Systems. Humana Press, 2001. 123-134.

Response: Yes. In this review, we only referenced and discussed the PODIP3 interactors that have been validated by additional assays, including immunoprecipitation in vitro and in vivo.

- In the reference [1], POLDIP3 is called PDIP38 (neither PDIP46 nor PDIP3). This nomenclature needs to be clarified. On the other hand, in which species does this protein receive the name POLDIP3?

Response: In the cited reference, PDIP46 is the POLDIP3, which is the official name for all species. The PDIP38 is a different protein, whose official name is POLDIP2 (https://www.ncbi.nlm.nih.gov/gene/26073).

- Line 41: the authors talk about "aspartic acid-55", while Figure 1 shows that it is located in position 53. Which is the correct one? On the other hand, to which species does the sequence referred to in Figure 1 belong?

Response: It should be arginine-53. We have corrected it in section-2.

- Line 47: what kind of modified RNA? mRNA? lncRNA? rRNA? tRNA? microRNA? Based on the reference given, it seems to be mRNA, but it should be clearly indicated in the text. On the other hand, what kind of modification is involved? Based on the reference given, it seems to be 5-methylcytosine. Where are these modifications located? UTR? CDS?

Response: we have clarified the modification in section-2

- It should be explained what the red rectangles in Figure 1A mean.

Response: We have added information related to the red rectangles in Fig 1A

- Figure 3 is excessively pixelated and the label "Fork repair and re-start" is cropped at the left border.

Response: It becomes the new Fig 5. We have fixed the issues related to it.

- It is necessary to include a phylogeny of POLDIP3 with other evolutionarily related proteins and discuss its evolutionary conservation in eukaryotes.

Response: we have created a phylogeny of POLDIP3 (as the new Fig 3)

- It is necessary to include the predicted 3D structure of POLDIP3. It can be found in the AlphaFold Protein Structure Database (https://alphafold.ebi.ac.uk/). The locations of functional domains and residues that may undergo modifications should be highlighted in this structure. I recommend using a different color for each functional domain of the protein and a color code according to the type of chemical modification that the residues may undergo.

Response: we have added the predicted 3D structure of POLDIP3 based on the AlphaFold Protein Structure Database (https://alphafold.ebi.ac.uk/) (as the new Fig 2).
